ecology

bee, crop, biogeography, plant–pollinator interaction, agriculture

**Author for correspondence:**
Julian Brown
e-mail: julian.macpherson.brown@gmail.com

# Global-scale drivers of crop visitor diversity and the historical development of agriculture

## Julian Brown and Saul A. Cunningham

Fenner School of Environment and Society, Australian National University, 141 Linnaeus Way, Acton, Australian Capital Territory 2601, Australia

 JB, 0000-0003-1443-251X

Understanding diversity in flower-visitor assemblages helps us improve pollination of crops and support better biodiversity conservation outcomes. Much recent research has focused on drivers of crop-visitor diversity operating over spatial scales from fields to landscapes, such as pesticide and habitat management, while drivers operating over larger scales of continents and biogeographic realms are virtually unknown. Flower and visitor traits influence attraction of pollinators to flowers, and evolve in the context of associations that can be ancient or recent. Plants that have been adopted into agriculture have been moved widely around the world and thereby exposed to new flower visitors. Remarkably little is known of the consequence of these historical patterns for present-day crop-visiting bee diversity. We analyse data from 317 studies of 27 crops worldwide and find that crops are visited by fewer bee genera outside their region of origin and outside their family's region of origin. Thus, recent human history and the deeper evolutionary history of crops and bees appear to be important determinants of flower-visitor diversity at large scales that constrain the levels of visitor diversity that can be influenced by field- and landscape-scale interventions.

## 1. Introduction

Flowers that attract insects have been a feature of most angiosperm lineages since the Cretaceous [1], and with the rise and diversification of angiosperms, they have become the dominant reproductive mode for terrestrial plant life, including those plants we have domesticated as crops. Because insect pollination is important to so many angiosperms [2], and pollinator taxa differ in the ways they interact with flowers, it has been argued that the diversity of angiosperm lineages and their many floral forms arises in part from differential selection driven by different pollinator taxa [3]. The emergence and diversification of bees in particular has been implicated in the rise and diversification of angiosperms [4], and bees are the primary pollinators for most pollinator-dependent crops [5,6].

The dependence of angiosperms on particular animals for reproduction could present a problem for plants dispersing to new geographical locations. Baker [7,8] inferred that specialized pollinators, as well as mates, may be absent from newly colonized areas, based on observations of self-compatibility being more common at the periphery of the ranges of several plant taxa. More recent studies of plant invasions confirm the advantage of self-compatibility for invasive plants, but conclude that pollinator shortage is less important than mate availability for self-incompatible invaders because generalist pollinators are generally found in the introduced range [9,10]. Similarly, those many crops that have successfully spread across a wide geographical range are evidence that effective pollinators have been encountered in a wide range of

places, though there is a common belief that the coincident introduction of generalist honeybees (*Apis mellifera*) underlies this success [11].

While pollinator-dependent crops yield valuable harvests even when cultivated away from their region of origin, some lines of evidence indicate that flower-visitor diversity influences agricultural outcomes nevertheless. Global reviews suggest that pollen limitation of fruit set is surprisingly frequent in crops [12], and that richer flower-visitor assemblages are associated with better crop pollination and fruit set [13,14].

Further to consequences for crop production, it is important to understand drivers of crop-visiting bee diversity in terms of the biodiversity conservation value of agricultural landscapes. Exotic plants have been shown to attract fewer pollinator taxa compared to native plants in one north American study [15]. If this pattern proves to be general, it would help to explain why typically only 12–13% of the bee species within a region are observed exploiting agricultural crops [16].

While the field-scale drivers of crop-visitor diversity, such as pesticide and habitat management, are reasonably well understood [17,18], drivers arising from larger-scale biogeographic patterns have not been examined. Three predictions regarding global-scale drivers of crop-visitor diversity are tested in this study, based on the following rationale.

## (a) Prediction 1: the home ground advantage
Geographical variation in crop-visitor diversity might arise from the biogeographic history of each crop species and the opportunities for plant and bee adaptation this entails. A parallel can be found in the enemy release hypothesis, which states that exotic species escape specialist enemies that have adapted to them in their native range [19]. Consistent with this, the number of herbivore, parasite, fungus and virus species attacking introduced plants and animals tends to be higher in their native ranges [20–22]. In a similar way, one might expect that plants attract more bee taxa to their flowers in their regions of origin where there has been time for selection to favour the beneficial association, and relatively fewer in places of recent introduction. This assumes plant lineages accumulate visiting bee taxa over evolutionary timescales relevant to bee speciation. On the other hand, the evolution of highly specialized pollination systems may exclude many bee taxa, leading to a narrowing of the pool of likely visitors. Some crops require moderately specialized forms of pollination, such as tomatoes and blueberries that are pollinated by the subset of bee taxa capable of vibrating anthers to release pollen [5,23]. However, we argue domestication of plant species into agriculture is unlikely to favour those with extreme floral specialization as this may lead to crop failures when crops are moved to new locations (e.g. oil palm [24]), and that therefore this phenomenon is likely to be rare among crops. We therefore predict that crops will attract more bee genera in their regions of origin compared to other realms.

## (b) Prediction 2: the benefit of being surrounded by relatives
The deeper biogeographic histories of crop lineages might give rise to another level of geographical variation in crop-visitor diversity, in which the presence of confamilial species influences pollinator availability. This idea is similar to the observation that specialist natural enemies native to the invaded range can be pre-adapted to exotic plants through adaptation to the native relatives of these plants in the invaded range [25]. Similar host switching has been observed in highly specialized bees, such as New World *Peponapis* (Cucurbitaceae specialists) and *Diadasia* (Asteraceae or Malvaceae specialists) that readily visit introduced Old World crops as long as they come from their preferred host families [26,27]. The conservation of pollinator-relevant traits, such as pollen quality, at the plant family level [28] presumably makes this possible. Again assuming that plant lineages accumulate bee taxa through time, we predict that plants will attract more bee taxa in regions where their family has had the longest evolutionary history with local bees, even if the focal crop species has its origins in a different region.

## (c) Prediction 3: diversity benefits accumulate across the suite of crop species
If crops are visited by more bee taxa in their and their family's realm of origin (predictions 1 and 2), one might expect the suite of native crops grown in a region to collectively support more bee taxa than are supported by the suite of exotic crops (where 'native' is used to indicate being in the crop's realm of origin or the crop family's realm of origin). This may not be the case, however, if the crops native to a particular region are dominated by species that support fewer bees in general (e.g. because they possess floral traits that exclude some bee taxa), despite supporting more bees in their native compared to introduced ranges. Since we argue that crops will typically exhibit generalized pollination systems, we predict that suites of native crops will support more bee taxa than suites of exotic crops grown in each region.

## 2. Material and methods

### (a) Design
To test these predictions, we analysed observations of flower-visiting bees for 27 crop species in six families (Asteraceae, Cucurbitaceae, Fabaceae, Malvaceae, Rosaceae and Solanaceae) (figure 1). Each plant family includes at least one crop species originating in the New World (i.e. Nearctic and Neotropics) and one in the Old World (i.e. Palaearctic, Afrotropics and Indomalaya, figure 1), with widespread geographical range in contemporary agriculture, and which exhibit some dependence on bee-mediated pollination to maximize seed set [29]. For a crop species to be included in this study required that there were observations of flower-visiting bees in its realm of origin and at least one other realm.

Having selected a candidate list of plant families, we conducted a literature review of bee visitation to the crop species in each family using Web of Science, CAB Abstracts and Google Scholar (first 100 hits only) with the following search terms: (genus AND species AND visit*) OR (genus AND species AND pollinat*), where genus and species included all synonyms for each crop. These searches returned over 10 000 articles, so a hierarchy of filters was applied to focus on the most relevant studies. The first filter was applied to titles of papers to remove obviously irrelevant material (e.g. studies comparing effectiveness of different crop varieties as pollinizers). A second filter applied to title and abstract for relevance (e.g. any mention of flower visitors or pollinators). A third filter was applied to

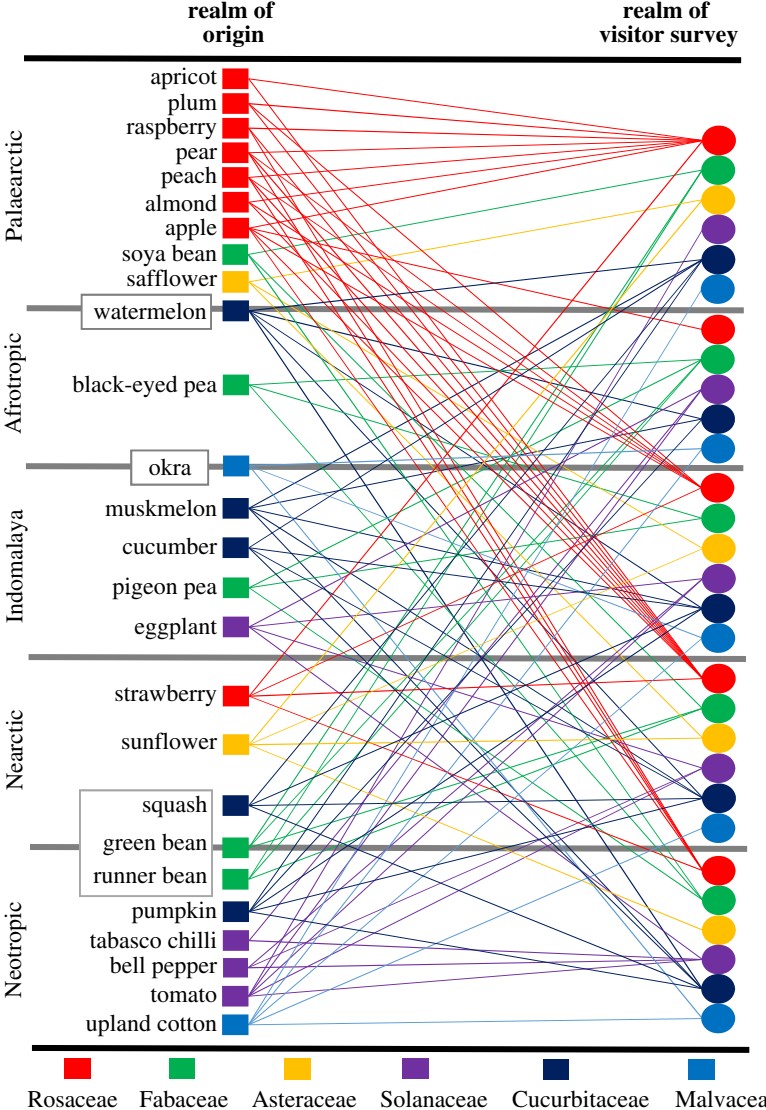

**Figure 1.** Biogeographic realm for the origin of each crop and locations where flower visitors have been surveyed. Crops with multiple origins or uncertainty regarding crop origin are placed on the appropriate boundary inside a box. Six different plant families are distinguished by different colours. Lines connect the crop origin (squares) to each realm in which visitors have been observed for that crop (circles). Common names used for crops apply to the following species: Asteraceae—*Carthamus tinctorius* (safflower), *Helianthus annuus* (sunflower); Cucurbitaceae—*Citrullus lanatus* (watermelon), *Cucumis melo* (muskmelon), *C. sativus* (cucumber), *Cucurbita moschata* (pumpkin), *C. pepo* (squash); Fabaceae—*Glycine max* (soya bean), *Vigna unguiculata* (black-eyed pea), *Cajanus cajan* (pigeon pea), *Phaseolus vulgaris* (green bean), *P. coccineus* (runner bean); Malvaceae—*Abelmoschus esculentus* (okra), *Gossypium hirsutum* (upland cotton); Rosaceae—*Prunus domestica* (plum), *P. armeniaca* (apricot), *P. dulcis* (almond), *P. avium* (sweet cherry), *P. persica* (peach), *Malus domestica* (apple), *Pyrus communis* (pear), *Rubus idaeus* (raspberry), *Fragaria X ananassa* (strawberry); Solanaceae—*Solanum melongena* (eggplant), *S. lycopersicum* (tomato), *Capsicum annuum* (bell pepper), *C. frutescens* (tabasco chilli). (Online version in colour.)

ensure appropriate taxonomic resolution of flower-visiting bees, selecting those studies that identified all bees visiting crops to at least genus level. Where these data were not included in the published article but it seemed likely the data had been collected, we contacted authors and requested these data. Two studies on small, isolated islands were excluded due to possible island biogeography effects.

We chose bee genus, rather than species, to reduce the risk of error in species identification and because genus reflects a deeper level of trait diversity among bees. Between-study differences in the number of bee genera reported could result from the fact that different studies were conducted in different decades encompassing different understandings of bee taxonomy, so genus names were compared to the online Integrated Taxonomic Information System (ITIS, https://www.itis.gov) and recent literature (any taxonomic revisions obtained from the first 10 hits on Google Scholar using the following search terms: genus AND taxonom*) when ITIS record reviews were greater than 5 years old.

The following data were then extracted from each study:

(1) number and identity of bee genera observed visiting each crop,
(2) biogeographic realm and latitude of location(s) where crop visitors were observed (centroid of locations when there were multiple within a study),
(3) number of locations where crop visitors were observed.

We recorded the numbers of locations observed in each study because the number of bee taxa recorded visiting any crop is likely to increase with the number of locations surveyed, particularly where surveys occur across gradients of land cover. We noted the latitude of the location (or the centroid where there were multiple locations) to account for underlying latitudinal gradients in species richness [30]. We used biogeographic realm [31] as the large-scale spatial grouping for testing the predictions in this study. This was used to determine whether each

crop in each study was being grown in its region of origin or family origin. The biogeographic realm of observation was also included as a predictor variable in modelling to account for any possible geographical bias in taxonomic understanding or survey effort. Whereas Takhtajan's floristic regions have been used to define regions of crop origins [32], we chose to use the biogeographic realm concept because it has fewer but larger regions, allowing a simpler study design with more comparisons per realm. Only five realms (Nearctic, Neotropic, Palaearctic, Afrotropic and Indomalay) were included in the analyses as there were too few published bee observations from other realms (Australasia, Oceania and Antarctica).

## (b) Crop biogeography

The biogeographic histories of crop species and plant families were determined on the basis of a literature search using Google Scholar with the following search terms: (genus AND species AND orig*) OR (genus AND species AND domesticat*), and (family AND orig*) or (family AND evolution). We attempted to attribute a single origin to each crop and family according to the most recent published studies, though some crops and families were given multiple origins in cases where origins are currently controversial or there were multiple, independent domestications in different biogeographic realms (electronic supplementary material, table S4). The descriptions are the most current found for family and crop origin based on best available data. Box 1 graphically illustrates an example history for one family, the Cucurbitaceae.

## (c) Statistical analysis

To provide insight into between-realm differences in overall richness of genera and differences in the amount of data available in our dataset, we provide accumulation curves for the number of bee genera observed in each realm (figure 2) using the sample-based rarefaction method 'exact' in the R package vegan [42]. We also provide tables listing the crop-visiting bee genera in each biogeographic realm (electronic supplementary material, tables S5 and S6).

Our first two predictions—that crops will be visited by more bee taxa in their realm of origin and family origin—were tested using regression models with each crop in each study treated as an observation. We modelled the number of bee genera visiting the crop as a function of the terms described in electronic supplementary material, table S1, with crop origin and family origin representing our first two hypotheses, and all other terms used to account for additional sources of variation in bee genus richness.

We used mixed-effects models with crop species as a random effect to account for possible intrinsic differences between crops with more or less generalized pollination systems. Poisson models were over-dispersed, so negative binomial models were used, with a quadratic rather than linear mean–variance relationship specified as it provided a better fit to the data. Generalized variance inflation factors were calculated for the global model (i.e. containing all variables listed in electronic supplementary material, table S1) and were all less than 1.9, suggesting that multicollinearity was not at a level that would complicate interpretation.

We used an information theoretic approach to model selection. We first used AICc to compare the global model with linear and quadratic functions of latitude as there is some evidence of a hump-shaped relationship between bee richness and latitude [30], but we found the model with the linear function to be substantially better (greater than 2 AICc), so included only the linear function in further modelling. We then used AICc to compare models containing all possible combinations of latitude, number of locations (log-transformed to linearize), crop origin,

family origin and biogeographic realm. All analysis was performed in R, using packages glmmTMB [43] for mixed-effects models and MuMIn [44] for model comparisons.

Thirty studies described visitors to more than one of the focal crops, which introduces some risk of non-independence. However, when we performed the analysis after removing studies with multiple crops, the model selection outcomes were unchanged, and parameter estimates and confidence intervals changed only slightly, so we present the analysis of data from all studies.

Our third prediction—that suites of crops native to a region collectively support more bee taxa compared to exotic crops—was evaluated by comparing bee genus accumulation curves for native crops and exotic crops within each biogeographic realm. The identity of each bee genus recorded visiting each crop across all studies within each biogeographic realm was used to produce crop species–bee genus interaction networks (i.e. presence/absence of each crop species–bee genus interaction) for each biogeographic realm. These networks were then used to produce for each realm two accumulation curves (with 95% confidence intervals) for the number of bee genera observed in the network as crop species were added, one curve with 'native' crops as samples and one with 'exotic' crops as samples. We defined 'native' crops as those originating in or belonging to families originating in the focal realm, and 'exotic' crops being those originating in and belonging to families originating in other realms. Accumulation curves were used to determine whether the number of bee genera entering each network increased more rapidly with the addition of native compared to exotic crops to the network, such that agricultural regions with a greater diversity of native compared to exotic crops would be expected to support more bee genera. The sample-based rarefaction method 'exact' in R package 'vegan' [42] was used to generate these curves as it is suitable for presence/absence data.

## 3. Results

## (a) Data

Our literature search produced a database of 317 articles (see electronic supplementary material, data S1) describing bee genera visiting 27 crop species across 1520 locations in five biogeographic realms (figures 1 and 2; electronic supplementary material, table S3). The number of crops surveyed was similar in all realms (20–23) except the Afrotropics (14) (figure 2). The number of studies was higher in the Indomalay (120) compared to all other realms (30–63) (figure 2). There was also between-realm variation in the distribution of survey effort between crop families. Rosaceae was the most or second most surveyed in terms of the number of crops and number of locations in all realms, except the Afrotropics where it was the least studied by both measures. Conversely, Fabaceae was the most well-studied family in the Afrotropics and relatively less studied elsewhere.

Forty-five bee genera were detected visiting crops in all biogeographic realms, including *Apis*, *Ceratina*, *Eucera* and *Xylocopa* from the Apidae family, *Halictus* and *Lasioglossum* (Halictidae), *Colletes* (Colletidae) and *Megachile* (Megachilidae) (electronic supplementary material, table S5). Most of these genera were detected visiting crops from most families in all realms (i.e. widespread generalists), whereas *Eucera* and *Colletes* were detected visiting crops from only one or two families in most realms (data not shown). Eighty-seven bee genera were observed visiting crops in only one realm (electronic supplementary material, table S6), and most of these

**Box 1.** Biogeographic history of crops in the Cucurbitaceae.

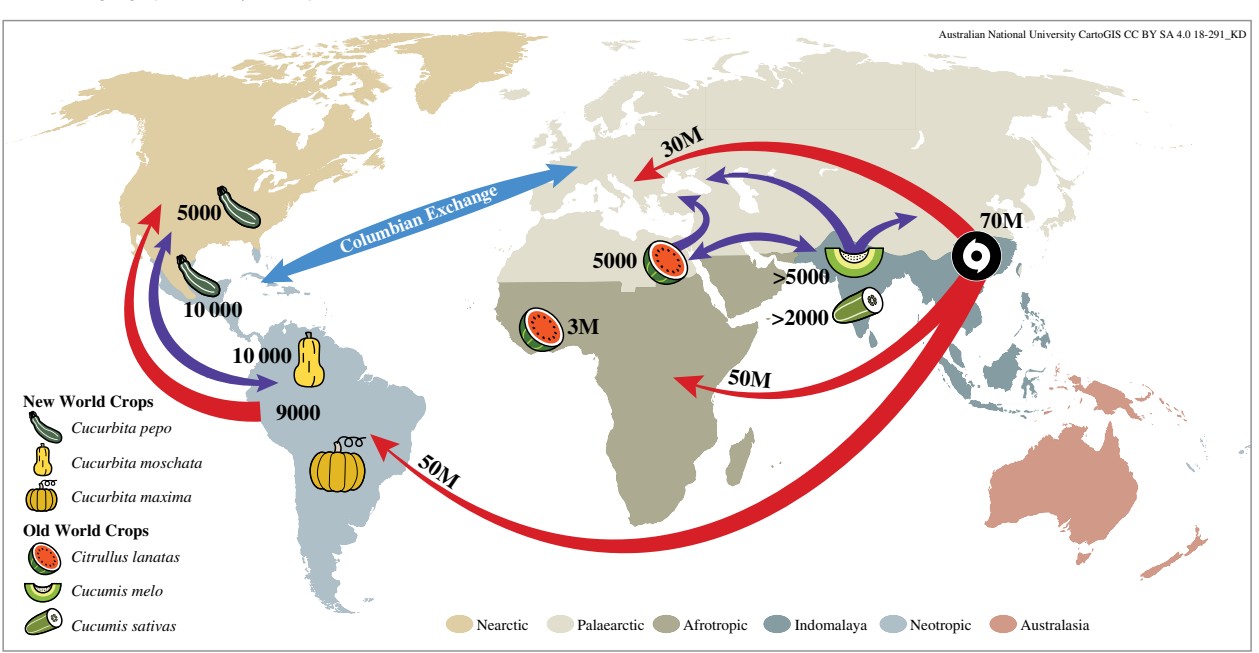

Australian National University CartoGIS CC BY SA 4.0 18-291_KD

Cucurbitaceae crops are used to illustrate the history of plant movement in three time periods: pre-human (greater than 10 000 years before present (ybp), red arrows), human-mediated pre-Columbian (10 000–500 ybp, purple arrows, double head indicates two-way dispersal) and human-mediated post-Columbian (less than 500 ybp, blue arrow). Note that for the earliest dates, the position of continents were different. The Cucurbitaceae family originated 70 million ybp in the area we now call Indomalaya, spread first to the Afrotropics and Neotropics and then to Palaearctic Europe and the Nearctic [33].

Cultivated *Cucumis melo* and *C. sativus* first appeared in the Palaearctic 5000 ybp and 2000 ybp, respectively [34], although they are thought to have been domesticated where their wild progenitors originated in Indomalaya [35]. *Citrullus lanatas* originated in sub-Saharan Africa around 3 million ybp [36], and was first cultivated there or in Palaearctic north Africa 5000 ybp [37]. It spread to Europe and the Indomalaya before the Columbian exchange [38]. *Cucurbita pepo* was domesticated twice independently, once along the Neotropic–Nearctic border around 10 000 ybp, and again further north in the Nearctic around 5000 ybp [39]. *Cucurbita moschata* was domesticated in the northern Neotropics around 10 000 ybp and *C. maxima* in the southern Neotropics sometime after [39,40]. There is evidence of crop exchanges between the Nearctic and Neotropics in pre-Columbian times [41], though archaeological *Cucurbita* remains have not typically been identified to species. All species attained global distributions following the arrival of Europeans in the Americas around 500 ybp.

were detecting visiting crops from only one or two families (data not shown).

Genus accumulation curves for each biogeographic realm revealed between-realm differences in overall richness of bee genera, and differences in the amount of data available (figure 2). Bee genera accumulated most rapidly and reached their highest number in the Neotropics (approx. 75) where the slope was relatively steep at the highest number of studies (i.e. still accumulating), indicating that true (cf. sampled) bee diversity is likely to be even higher. By contrast, the Indomalay curve virtually plateaued at a relatively low number of bee genera (approx. 35). The Palaearctic curve was similar to Indomalay, the Nearctic somewhere between Palaearctic and Neotropics, while the Afrotropics curve exhibited a slope almost as steep as the Neotropics though reaching only 40 genera due to the smaller number of studies from this realm. Thus, the true number of crop-visiting bee genera is likely to be higher in the New compared to Old World, though the Afrotropical realm was less sampled, and whether this reflects actual genus diversity or geographical differences in taxonomic understanding is unknown.

## (b) Regression analysis

Our first two predictions were supported by results of regression analyses, with geographical variation in crop-visitor diversity explained by the biogeographic histories at the level of crop and also crop family. The top-ranked regression models explaining variation in the number of crop-visiting bee genera contained crop origin and family origin, as well as biogeographic realm, latitude and number of locations (electronic supplementary material, table S2). The top three models encompassed greater than 95% of the Akaike weight and all contained crop origin, biogeographic realm and number of locations, though only two contained family origin and latitude, indicating these two variables were less important (electronic supplementary material, table S2). According to the best model, which was substantially better than all other models ($\Delta$AIC > 2), 95% confidence intervals excluded one for all parameter estimates (figure 3). There is currently no accepted method for calculating explained variance for negative binomial models (nbinom2) in glmmTMB, so we used a generalized linear

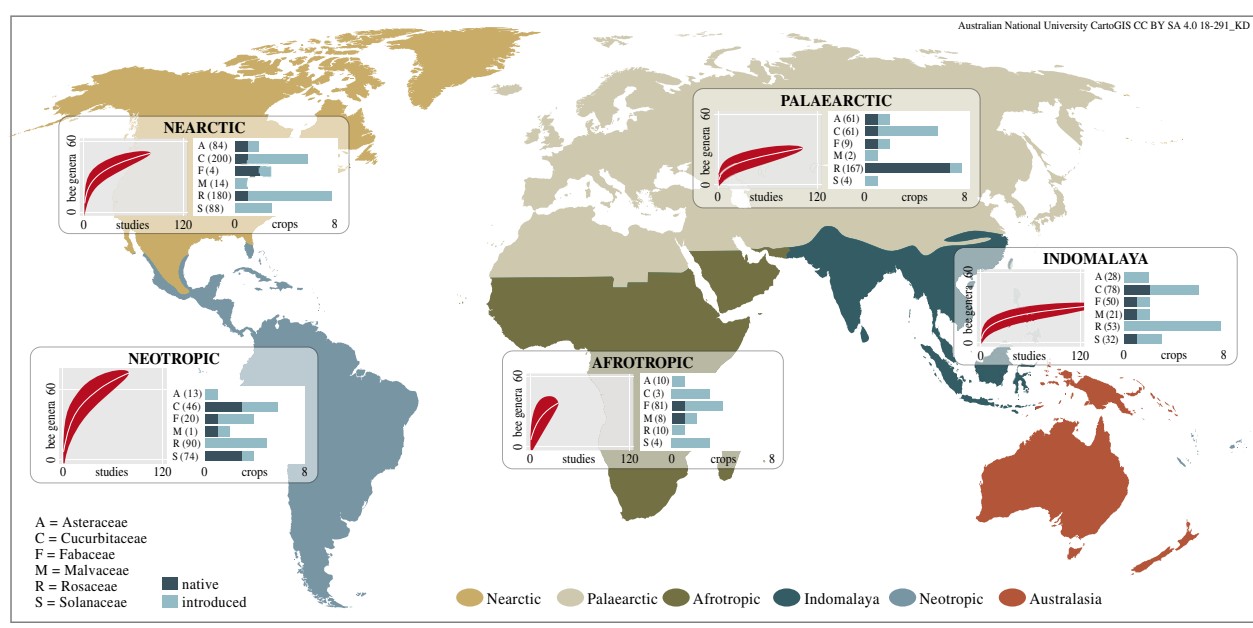

**Figure 2.** Bee and crop diversity surveyed across five biogeographic realms. Bee genera accumulation curves show diversity and depth of sampling in each realm. Bar graphs indicate the number of native and introduced crops studied in each plant family, and the total number of locations at which crops from each family were studied (in brackets). (Online version in colour.)

model identical to the full model but with crop identity as a fixed effect to calculate deviance squared (analogous to $R^2$ for generalized linear models), which equalled 49%.

In agreement with our first prediction, crops were visited by more bee genera in their biogeographic realm of origin compared to other realms. The average number of bee genera visiting crops across all studies of a single location in their regions of origin was 4.5. Crops grown outside their realm of origin attracted 1.3, or approximately 27%, fewer genera on average (figure 3). The second prediction was also supported by the data and had a similar effect size, as crops were visited by 1.3 more bee genera when grown in the realm where their family originated than when grown outside this realm (figure 3).

To demonstrate that the effect of crop origin is reciprocal, i.e. that the same patterns apply for introductions from New to Old World and Old to New world, we produced electronic supplementary material, figure S3 using the best regression model (above) but with crop species treated as a fixed effect, and an interaction between crop and crop origin. Twenty of the 27 crops (74%) showed some indication of greater crop-visitor diversity when grown in the region of origin, with similar numbers of these crops originating in the New World ($n = 9$) and Old World ($n = 11$). This shows that effects are reciprocal.

We also found that the number of bee genera visiting crops declined with increasing latitude of study location (i.e. moving from the equator to the poles), increased with the number of sites, and was higher in the Nearctic and Neotropic compared to all other realms (figure 3).

### (c) Genus accumulation curves
Our third prediction found mixed support from genus accumulation curves. Since regression modelling found positive, additive effects of crop origin and family origin on crop-visiting bee genera, we compared bee genus accumulation curves for both native (grown in realm of crop origin

or family origin) and exotic crops. In the predominantly tropical realms (Neotropics, Afrotropics and Indomalaya), bee genera accumulated in networks more rapidly for native compared to exotic crops (figure 4). In other words, for a given number of crop species in the network, the number of bee genera in the network tended to be higher when crops were native, though 95% confidence intervals generally overlapped until 6–7 crops species were reached. This pattern did not hold for the predominantly temperate Nearctic and Palaearctic realms, where bee genera accumulated at a similar rate for native and exotic crops (figure 4).

## 4. Discussion
Recently observed declines of insect pollinators [45,46] have motivated much research into the drivers of crop-visitor diversity, which has revealed important factors such as habitat and pesticide management that operate over spatial scales from fields to landscapes [17,18]. This study shows for the first time the influence of drivers of diversity that operate at the global scale, and which ultimately constrain the levels of crop-visitor diversity that field- and landscape-scale interventions work with. Specifically, we find that the number of bee genera visiting crops is reduced when crops are grown (i) outside their region of origin or their family's region of origin, (ii) at higher latitudes, and (iii) in the Old World realms.

Our findings suggest that present-day global variation in crop-visitor diversity arises from the particular biogeographic history of each crop. Crop species tended to attract more bee genera in their realm of origin, where there has been more time for adaptation to occur. Most crop introductions to biogeographic realms outside regions of origin have occurred in the last 500 years, during exchanges of organisms between the New and Old World realms following the arrival of Europeans in the Americas. While some phytophagous insects have adapted to plants introduced during this time [25], our results suggest the diversification of bees has not been

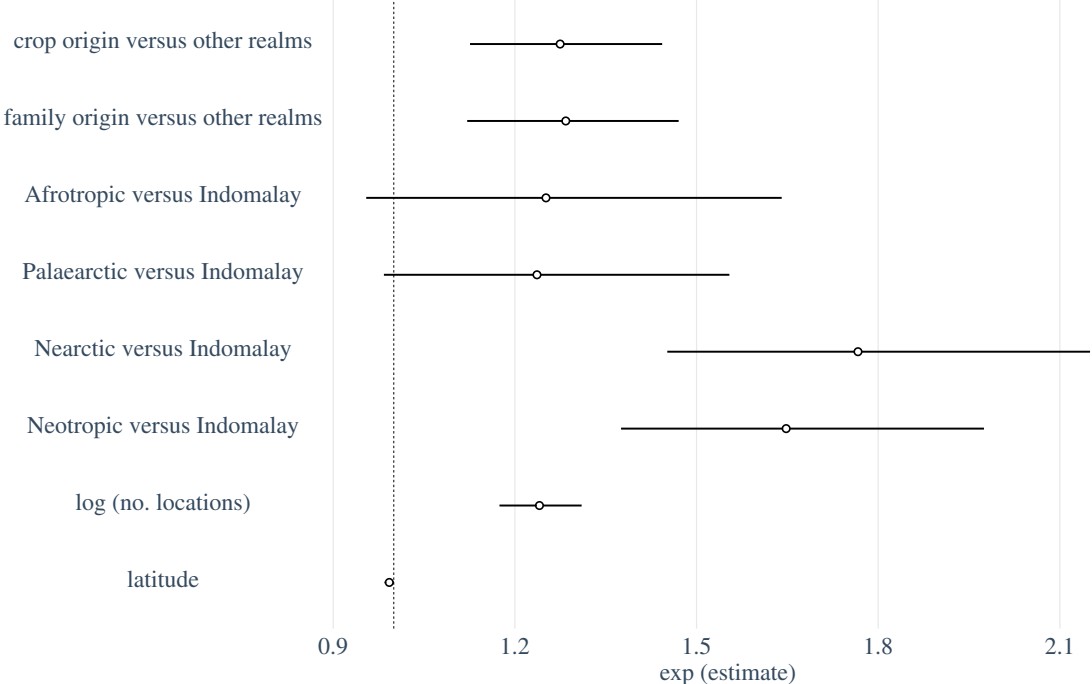

**Figure 3.** Estimated increase in the number of bee genera visiting crops when grown in the realm of crop origin and realm of crop family origin compared to all other realms, as well as Afrotropic, Palaearctic, Nearctic and Neotropic realms compared to the Indomalayan realm (Indomalay was used as reference as it exhibited the lowest average number of crop-visiting bee genera). Parameter estimates are derived from the top-ranked regression model (electronic supplementary material, table S2), with 95% confidence intervals, back-transformed (exponentiated) for ease of interpretation. Note that effects of predictors on the response are multiplicative (not additive) as a log-link function is used in negative binomial models, such that estimates with confidence intervals overlapping 1 (dashed line) are considered non-significant (at $\alpha = 0.05$).

sufficiently rapid to compensate the loss of crop visitors from regions of origin. While this might be seen as an argument for the co-introduction of exotic pollinators from regions of exotic crop origins, the negative ecological effects of introducing bees often outweigh positive effects [47].

We also found that crops attracted more bee genera in the realm of their family's origin. This suggests some level of pre-adaptation to exotic relatives of native mutualistic partners, which is consistent with observations of highly specialized bees visiting exotic plants belonging to their native host families [26]. However, it is not necessarily the case that all individuals within a bee species are sufficiently pre-adapted to adopt an exotic crop, as there can be intra-specific variation in the capacities of bees to use particular floral host taxa [48]. Intra-specific variation in the adoption of exotic crops is worth further investigation as a potential source of anthropogenic speciation driven by the adaptation of native insect (sub)populations to introduced plants [25].

Knowledge of crop biogeographic history can be used to refine biodiversity conservation strategies in agricultural landscapes. Current practices that are effective in promoting bee diversity in production landscapes are the retention or restoration of non-farm vegetation, reduced pesticide use and increased on-farm floral diversity through wildflower plantings or crop polycultures [16,17]. Our results suggest the species composition of crops could also influence bee biodiversity. Based on the current understanding of the biogeographic histories of the 27 crops we sampled, more bee genera were supported in the Neotropics, Indomalaya and Afrotropics by native crops compared to exotic crops. Since closely related bees tend to visit closely related plants [49], native crop assemblages might support more

bee genera if they are more taxonomically diverse, but this did not appear to explain our results (see electronic supplementary material). Thus, agricultural landscapes dominated by native crops are expected to support greater bee genus diversity in the predominantly tropical biogeographic realms.

By contrast, native crops did not collectively support more bee genera compared to exotic crops in the predominantly temperate Nearctic and Palaearctic realms. The native crops of temperate realms might exhibit floral traits tending to support fewer of the bee taxa favoured by agricultural landscapes. All Rosaceae crops are native to the temperate realms and are grown in seasonally cool environments at higher latitudes or altitudes where they typically flower for a short period of time in early spring. While many crops native to the predominantly tropical realms can be grown in the same seasonally cool environments, they typically flower later in spring or summer [50–52]. In the seasonally cool northeast of the USA, where crops from all biogeographic realms are grown, bee richness peaks during early spring in native forests, but peaks during summer in agricultural landscapes due to compositional shifts from early spring-active to summer-active bee taxa [53]. If agriculture generally favours summer-active bees over early spring-active bees in seasonally cool environments, native crop assemblages of temperate realms characterized by many Rosaceae and other early spring flowering taxa (e.g. blueberry and cranberry: Ericaceae) may generally support fewer of the bee taxa associated with agricultural landscapes.

More bee genera were recorded visiting crops in the New World (Nearctic and Neotropic) rather than Old World realms, but there are multiple possible contributors to this

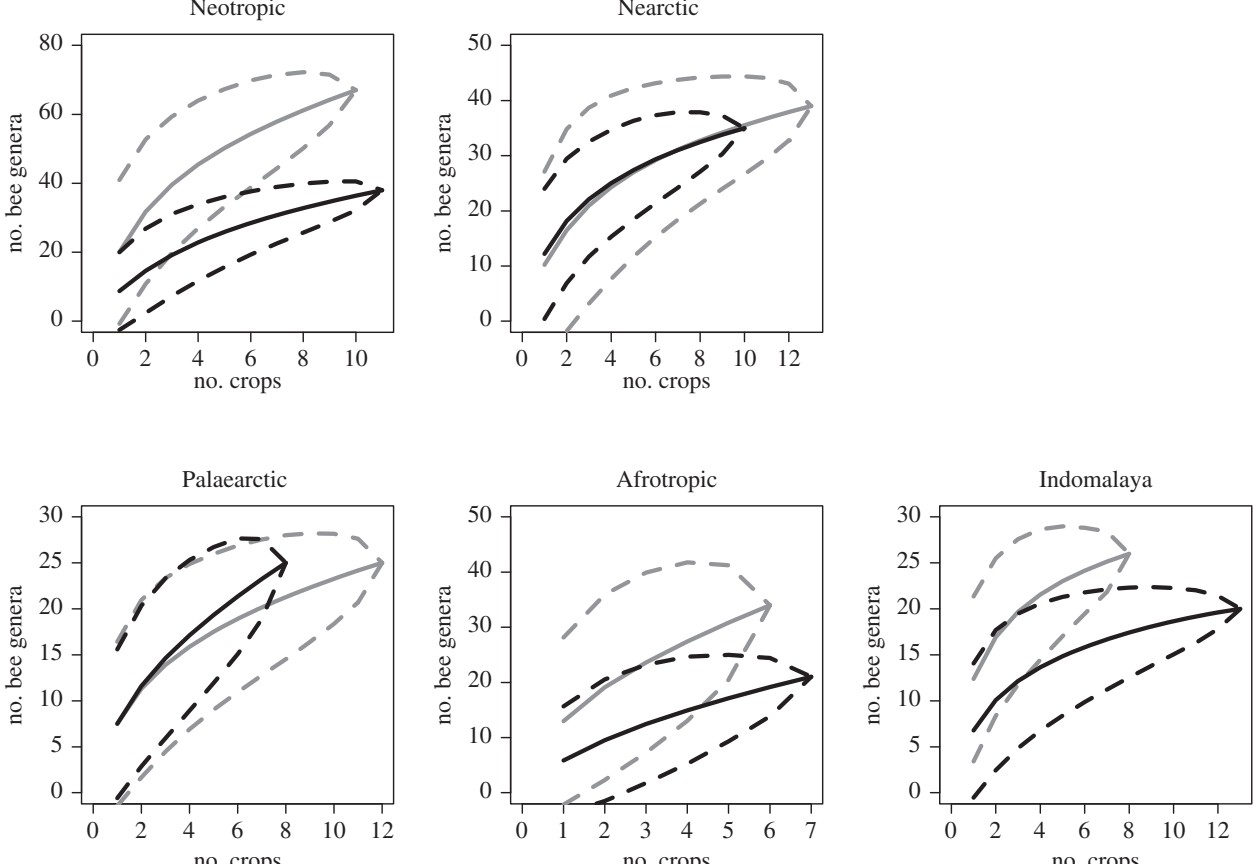

**Figure 4.** Bee genus accumulation curves for each biogeographic realm. The *y*-axis shows the number of bee genera observed on flowers and the *x*-axis shows the number of crops studied. Solid grey lines are accumulation curves when native crops are sampled (grey dashed lines are 95% confidence bands), solid black lines are accumulation curves when exotic crops are sampled (black dashed lines are 95% confidence bands).

apparent diversity pattern. There could be a spurious influence from geographical differences in taxonomic practice. This pattern could also be influenced by between-realm differences in cropping systems, since New World realms are dominated by large-scale monocultures whereas most of Asia and Africa is dominated by smallholder farming [54], though monocultures are unlikely to support greater bee diversity [55], so the expected direction of effect is counter to the pattern we found. Crops might also attract more bee genera in the New World realms because the New World contains more bee taxa with traits that make them tolerant of agricultural environments. Tolerance of agriculture is greatest for bees that are social, soil nesting and happen to preferentially forage (specialize) on the plant taxa that we have co-opted into agriculture [53,56]. There are broad geographical patterns in the distribution of nesting, social and diet traits, though systematic assessments are lacking. Sociality is concentrated in the Meliponini which are most diverse in the Neotropics [57], and many of the well-known dietary specialist groups are concentrated in the New World [58,59] where they are found visiting crops (e.g. *Diadasia* and Panurginae genera *Acamptopeum*, *Calliopsis*, *Callonychiumt*, *Anthrenoides*, *Pseudopanurgus*, *Psaenythia* and *Perdita*; electronic supplementary material, tables S5 and S6). By contrast, stem-nesting (the most frequent alternative to soil nesting) is most frequent in Megachilidae and Xylocopinae which are least diverse in the New World. Further research is required to test the 'agricultural tolerance' hypothesis as our literature-based analysis was unable to distinguish biological from non-biological explanations of biogeographic realm effects.

The global-scale drivers revealed in this study indicate that human impacts on crop-visitor diversity have been occurring over thousands of years of inter-continental trade and migration, intensifying after the establishment of trade networks between New and Old World realms, and well before the advent of pesticides and agricultural intensification. Our findings of lower crop-visitor diversity when grown outside crop or family origins parallel observations of plant–enemy interactions [22,25]. The signal of evolutionary history thus appears in the era of globalization as constraints on novel interactions between plants, enemies and mutualists that determine the capacity of wild species to adapt to the agricultural landscapes we create.

**Data accessibility.** The data supporting the results are archived in the Dryad Digital Repository: https://doi.org/10.5061/dryad.np5hqbzp5 [60].

**Authors' contributions.** Study conceptualization and design by S.A.C. and J.B.; data collection and analysis by J.B.; writing led by J.B. with support from S.A.C.

**Competing interests.** We declare we have no competing interests.

**Funding.** Funding was provided by the AgriFutures 'Securing Pollination for More Productive Agriculture' project (grant no. PRJ-010561), supported by the Australian Government Department of Agriculture as part of its Rural R&D for Profit Program, and Horticulture Innovation Australia.

**Acknowledgements.** We thank Lucas Garibaldi, Simon Haberle and Jamie Stavert for comments on an early draft. Wade Blanchard provided advice regarding elements of the statistical analysis.

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
