## [Reviewer comments · Proceedings of the Royal Society B: Biological Sciences]

Review History

RSPB-2019-1463.R0 (Original submission)

Review form: Reviewer 1 (Stephen L. Buchmann)

Recommendation

Accept as is

Scientific importance: Is the manuscript an original and important contribution to its field?

Excellent

General interest: Is the paper of sufficient general interest?

Excellent

Quality of the paper: Is the overall quality of the paper suitable?

Excellent

Is the length of the paper justified?

Yes

Should the paper be seen by a specialist statistical reviewer?

No

Do you have any concerns about statistical analyses in this paper? If so, please specify them explicitly in your report.

No

It is a condition of publication that authors make their supporting data, code and materials available - either as supplementary material or hosted in an external repository. Please rate, if applicable, the supporting data on the following criteria.

Is it accessible?

Yes

Is it clear?

Yes

Is it adequate?

Yes

Do you have any ethical concerns with this paper?

No

Comments to the Author

I see no mention of buzz pollination by specialist bees (bumble bees etc.) in angiosperm crops having poricidal anthers. Tomato is an example. Honey bees are incapable of sonicating tomatoes and other similar crops (blueberry, cranberry, chile peppers, eggplant, kiwi fruit). This has been reviewed by S. Buchmann and others.

One issue I have is that the 317 papers only contained data on 27 crops plants around the world. That seems a bit odd. I would like to see the authors mention that there are often specialist bees on certain crop plants. They barely mention this fact. For example, in the New World, the genus Cucurbita is pollinated by a guild of specialist bees in the genera Xenoglossa and Peponapis. These associations originated in Mexico and the bees followed these domesticated crops as they moved north out of Mexico. Similarly, the bee genus Diadasia specializes on mallows, cacti and sunflowers. The sunflower bee Diadasia enavata in California is an example of such specialization.

You seemed to have missed citing the ground-breaking two volume book edited by David Roubik (STRI) and published by the FAO. It was published recently. In an earlier version, Roubik edited an FAO volume that had a table of crops grown around the world and listed their pollinators. It listed 1400 crop plant species. I am not sure why the articles you found only represent 27 crops. That seems very low. This might require a few sentences of explanation in the text. Please, please cite this ground-breaking work. I have attached a pdf file of the first volume for your consideration.

Congratulations on a fine paper. I especially like Figure 2.

Review form: Reviewer 2

Recommendation

Major revision is needed (please make suggestions in comments)

Scientific importance: Is the manuscript an original and important contribution to its field?

Acceptable

General interest: Is the paper of sufficient general interest?

Acceptable

Quality of the paper: Is the overall quality of the paper suitable?

Marginal

Is the length of the paper justified?

Yes

Should the paper be seen by a specialist statistical reviewer?

Yes

Do you have any concerns about statistical analyses in this paper? If so, please specify them explicitly in your report.

Yes

It is a condition of publication that authors make their supporting data, code and materials available - either as supplementary material or hosted in an external repository. Please rate, if applicable, the supporting data on the following criteria.

Is it accessible?

N/A

Is it clear?

N/A

Is it adequate?

No

Do you have any ethical concerns with this paper?

No

Comments to the Author

This manuscript explores how the diversity of pollinator services (bee genera) to crops change when crops are grown outside of the range they (or their progenitors or family members) are native to. While I found this a provocative and interesting question, I found the paper fell short of meeting its goals. One of the main concerns is that the sampling may be inadequate to answer the question and the presentation of the data and analyses lacked the transparency and specificity that would allow the reader to assess it.

I detail my main concerns here:

1. Although the paper's focus is bee genera, the names of these are never presented. Because the differences are relatively small (1 vs 4 genera) I was left wondering if the 1 was *Apis* thus reflected only visitation by cultivated species in many nonnative settings. The breadth and depth of the bee genera needs to be much better explicated.
2. The assessment of power (accumulation curves) came second, after the main regression

analyses. This seems backwards to me as I would imagine readers would appreciate understanding the rigor of the data first. I also thought that while the authors attempted to determine whether the number of crops sampled was enough, it was also important to know whether the sampling at a given crop was sufficient to capture all the diversity of visiting bees at that crop. The effort or intensity of pollinator sampling at this level was not addressed. It is possible that many bee taxa were missed but it is not possible to know with the data presented.

3. The presentation of the (regression coefficients, etc) statistical analyses was lacking from the main text and the details were not made clear in the supplement. I found this a serious concern for a reader attempting to assess the importance of any claim made in the paper.

4. I wondered about aspects of cropping systems that could affect pollinator diversity that differ in different parts of the world – large monoculture ag versus smaller or inter cropping systems. This should be addressed.

5. The assemblage effects section (starting line 290) was not mentioned earlier and it was not clear what 'networks' were being compared.

6. Figure 3 was not very helpful. Figures that described the bee taxa and their distributions would be much more helpful.

Decision letter (RSPB-2019-1463.R0)

05-Aug-2019

Dear Dr Brown:

I am writing to inform you that your manuscript RSPB-2019-1463 entitled "Global scale drivers of crop visitor diversity and the historical development of agriculture" has, in its current form, been rejected for publication in Proceedings B.

This action has been taken on the advice of referees, who have recommended that substantial revisions are necessary. With this in mind we would be happy to consider a resubmission, provided the comments of the referees are fully addressed. However please note that this is not a provisional acceptance.

To upload a resubmitted manuscript, log into <http://mc.manuscriptcentral.com/prsb> and enter your Author Centre, where you will find your manuscript title listed under "Manuscripts with

Decisions." Under "Actions," click on "Create a Resubmission." Please be sure to indicate in your cover letter that it is a resubmission, and supply the previous reference number.

Sincerely,
Dr Daniel Costa
mailto: proceedingsb@royalsociety.org

Associate Editor
Board Member: 1
Comments to Author:

This paper was reviewed by two experts and they reached quite different conclusions. Reviewer 1 one was positive, but felt that your scholarship in terms of citation of appropriate papers could be improved. Reviewer 2 was much more critical and identified some potentially serious flaws in the paper. The second reviewer felt that the data may be inadequate and that the methods and data handling were not sufficiently transparent. I agree with this reviewer about the backwards presentation of the accumulation curves. These curves should be used to assess whether the reported bee diversity in different regions is based on sufficient sampling. Cursory sampling of a crop outside its region of origin would lead to incorrect conclusions if contrasted with thorough sampling in the centre of origin. To be fair, you did include number of sampling localities in the models, but this is not really the same as using accumulation curves to assess whether sampling was sufficient or not. I also agree that Fig 3 is very poorly designed. I suggest putting Original realm and Introduced realm on the x axis and showing the actual means and errors connected by lines. The confidence intervals for the estimates here are perilously close to 0, suggesting that the effects are only just significant.

I felt that you need to place this study in the broader context of what happens to pollination systems when species are introduced to new regions. There is a whole body of theory going back to Herbert Baker and others that is completely missed here. From line 62 you relate your question to the enemy release hypothesis, but you miss out on the rich literature about pollination of plants introduced outside their native ranges, especially Baker's argument that this problem can be so severe that plants can only colonize new regions if they are capable of uniparental reproduction. To be sure Baker was mainly concerned with mate limitation, but he and others (notably Grant) also raised issues about limited pollinator availability outside the native range. The artificial selection that favours generalized pollination systems in crops is probably similar to the natural selection that favours generalized pollination systems in colonizing species. I would suggest consulting the literature on pollination of invasive plants, such as the reviews by David Richardson, Mark van Kleunen, John Pannell and others.

The Introduction would be more effective if it identified some specific predictions at the end after giving some prior rationale. In any case your two headings for rationale seem rather synonymous- "A home ground advantage" seems rather too similar to "The benefit of staying close to the family home". Perhaps you should make it clearer that you mean family in the taxonomic sense. Change to something like "the benefit of being surrounded by relatives"? Line 269 Please formally report the % of crops that show this effect (a decrease in bee diversity outside their native range) and break this down by realm. It is conceivable for example that if most crops originated in the Americas and that this region has a high bee diversity and that many are grown outside this region in areas of low bee diversity that the whole study is simply an artifact of high bee diversity in the place where most crops originate and not a general phenomenon. Can you show that effects are reciprocal, ie that the same patterns apply for introductions from New to Old World and Old to New world? By giving the % of crops that show the effect, the reader could judge whether or not the overall effect is due to a few crops that show pronounced differences.

Other comments

Lines 132 -189 This section reads like a list and should be placed in a table, either in the main section or supplementary

Line 195 You chose to analyse bee genera, but there seems no reason not to also analyse bee species richness or other higher level groupings.

Line 205 How did you choose this particular mean-variance relationship. I assume you mean to control for overdispersion in a Poisson model or do you mean that the quadratic term best accounted for overdispersion in the default negative binomial model?.

Referencing format needs to be updated to the Proc B style. This also applies to the boxes and some other elements as well.

Line 47 Make it clear that you are referring to crops and not plants in general

Lines 50-53 This sentence is incomprehensible and needs to be re-written. What does this value of 12-13% refer to? In a place such as Europe is there a non-agricultural landscape? Support in what sense – forage or nesting sites or both?

Line 274 Sentence incomplete – more bee genera when.....family originated than when grown outside this realm

Line 275 reword the number of bee genera visiting crops within studies declined. Delete “within studies” throughout the sentence

Fig 4 should be modified so that it also makes sense in greyscale.

Reviewer(s)' Comments to Author:

Referee: 1

Comments to the Author(s)

I see no mention of buzz pollination by specialist bees (bumble bees etc.) in angiosperm crops having poricidal anthers. Tomato is an example. Honey bees are incapable of sonicating tomatoes and other similar crops (blueberry, cranberry, chile peppers, eggplant, kiwi fruit). This has been reviewed by S. Buchmann and others.

One issue I have is that the 317 papers only contained data on 27 crops plants around the world. That seems a bit odd. I would like to see the authors mention that there are often specialist bees on certain crop plants. They barely mention this fact. For example, in the New World, the genus *Cucurbita* is pollinated by a guild of specialist bees in the genera *Xenoglossa* and *Peponapis*. These associations originated in Mexico and the bees followed these domesticated crops as they moved north out of Mexico. Similarly, the bee genus *Diadasia* specializes on mallows, cacti and sunflowers. The sunflower bee *Diadasia enavata* in California is an example of such specialization.

You seemed to have missed citing the ground-breaking two volume book edited by David Roubik (STRI) and published by the FAO. It was published recently. In an earlier version, Roubik edited an FAO volume that had a table of crops grown around the world and listed their pollinators. It listed 1400 crop plant species. I am not sure why the articles you found only represent 27 crops. That seems very low. This might require a few sentences of explanation in the text. Please, please cite this ground-breaking work. I have attached a pdf file of the first volume for your consideration.

Congratulations on a fine paper. I especially like Figure 2.

Referee: 2

Comments to the Author(s)

This manuscript explores how the diversity of pollinator services (bee genera) to crops change when crops are grown outside of the range they (or their progenitors or family members) are native to. While I found this a provocative and interesting question, I found the paper fell short of meeting its goals. One of the main concerns is that the sampling may be inadequate to answer the question and the presentation of the data and analyses lacked the transparency and specificity that would allow the reader to assess it.

I detail my main concerns here:

1. Although the paper's focus is bee genera, the names of these are never presented. Because the differences are relatively small (1 vs 4 genera) I was left wondering if the 1 was *Apis* thus reflected only visitation by cultivated species in many nonnative settings. The breadth and depth of the bee genera needs to be much better explicated.
2. The assessment of power (accumulation curves) came second, after the main regression analyses. This seems backwards to me as I would imagine readers would appreciate understanding the rigor of the data first. I also thought that while the authors attempted to determine whether the number of crops sampled was enough, it was also important to know whether the sampling at a given crop was sufficient to capture all the diversity of visiting bees at that crop. The effort or intensity of pollinator sampling at this level was not addressed. It is possible that many bee taxa were missed but it is not possible to know with the data presented.
3. The presentation of the (regression coefficients, etc) statistical analyses was lacking from the main text and the details were not made clear in the supplement. I found this a serious concern for a reader attempting to assess the importance of any claim made in the paper.
4. I wondered about aspects of cropping systems that could affect pollinator diversity that differ in different parts of the world – large monoculture ag versus smaller or inter cropping systems. This should be addressed.
5. The assemblage effects section (starting line 290) was not mentioned earlier and it was not clear what 'networks' were being compared.
6. Figure 3 was not very helpful. Figures that described the bee taxa and their distributions would be much more helpful.

Author's Response to Decision Letter for (RSPB-2019-1463.R0)

See Appendix A.

RSPB-2019-2096.R0

Review form: Reviewer 1

Recommendation

Accept with minor revision (please list in comments)

Scientific importance: Is the manuscript an original and important contribution to its field?
Excellent

General interest: Is the paper of sufficient general interest?

Excellent

Quality of the paper: Is the overall quality of the paper suitable?

Good

Is the length of the paper justified?

No

Should the paper be seen by a specialist statistical reviewer?

No

Do you have any concerns about statistical analyses in this paper? If so, please specify them explicitly in your report.

No

It is a condition of publication that authors make their supporting data, code and materials available - either as supplementary material or hosted in an external repository. Please rate, if applicable, the supporting data on the following criteria.

Is it accessible?

Yes

Is it clear?

Yes

Is it adequate?

Yes

Do you have any ethical concerns with this paper?

No

Comments to the Author

A fine paper and a topic that has not been addressed. I especially like Box 1 and Figure 1. I would, however, be happier if you can find a way to cut out a few pages of text, or move some "bulky" items to an electronic supplemental appendix. Otherwise, a great paper. Look forward to seeing it in print.

Review form: Reviewer 3

Recommendation

Accept with minor revision (please list in comments)

Scientific importance: Is the manuscript an original and important contribution to its field?

Excellent

General interest: Is the paper of sufficient general interest?

Good

Quality of the paper: Is the overall quality of the paper suitable?

Excellent

Is the length of the paper justified?

Yes

Should the paper be seen by a specialist statistical reviewer?

Yes

Do you have any concerns about statistical analyses in this paper? If so, please specify them explicitly in your report.

No

It is a condition of publication that authors make their supporting data, code and materials available - either as supplementary material or hosted in an external repository. Please rate, if applicable, the supporting data on the following criteria.

Is it accessible?

N/A

Is it clear?

N/A

Is it adequate?

N/A

Do you have any ethical concerns with this paper?

No

Comments to the Author

I think that the paper addresses a really interesting question in a thorough and systematic way. The previous round of reviews was very thorough and pointed out some significant limitations to the presentation of the work and how it was analysed. The manuscript as it stands looks fine to me and I can see that the authors have responded to the comments in detail. I therefore have no further requirements for changes, except that I spotted two typos:

Line 107 - "possess"

Line 251 - "actual"

I think that this has the potential to be a very well cited paper in the future as it brings together a fundamental idea in plant evolutionary ecology (Baker's Rule) with invasion biology and agricultural pollination biology. The authors are to be congratulated.

Decision letter (RSPB-2019-2096.R0)

21-Oct-2019

Dear Dr Brown

I am pleased to inform you that your manuscript RSPB-2019-2096 entitled "Global scale drivers of

crop visitor diversity and the historical development of agriculture" has been accepted for publication in Proceedings B.

The referee(s) have recommended publication, but also suggest some minor revisions to your manuscript. Therefore, I invite you to respond to the referee(s)' comments and revise your manuscript. Because the schedule for publication is very tight, it is a condition of publication that you submit the revised version of your manuscript within 7 days. If you do not think you will be able to meet this date please let us know.

In order to ensure effective and robust dissemination and appropriate credit to authors the

dataset(s) used should be fully cited. To ensure archived data are available to readers, authors should include a 'data accessibility' section immediately after the acknowledgements section. This should list the database and accession number for all data from the article that has been made publicly available, for instance:

Sincerely,

Dr Daniel Costa
mailto: proceedingsb@royalsociety.org

Associate Editor
Comments to Author:
Dear Dr Brown

I appreciated the clarity and thoughtful tone of your response letter and the attention to detail (including comments) in the revised manuscript. I feel that you have adequately addressed all of the issues raised by the original reviewers (and the few concerns that I had). The reviews of the revised version were also very positive. There are some minor spelling issues identified by the reviewers and I also noticed that some of your graphics have pixelated text and lines which suggests that the files were saved as full bitmaps and not as vector-based .eps files with editable vector-based graphical elements. Please make sure that all line and text components in the graphics are in a suitable vector format. Thank you for submitting this interesting global scale analysis to Proceedings B.

Yours sincerely
Steve Johnson

Editorial Office comments:

Please upload your figures as vector-based graphics in eps format to avoid fuzzy graphics at the print stage.

Reviewer(s)' Comments to Author:

Referee: 1

Comments to the Author(s).

A fine paper and a topic that has not been addressed. I especially like Box 1 and Figure 1. I would, however, be happier if you can find a way to cut out a few pages of text, or move some "bulky" items to an electronic supplemental appendix. Otherwise, a great paper. Look forward to seeing it in print.

Referee: 3

Comments to the Author(s).

I think that the paper addresses a really interesting question in a thorough and systematic way. The previous round of reviews was very thorough and pointed out some significant limitations to the presentation of the work and how it was analysed. The manuscript as it stands looks fine to me and I can see that the authors have responded to the comments in detail. I therefore have no further requirements for changes, except that I spotted two typos:

Line 107 – “possess”

Line 251 – “actual”

I think that this has the potential to be a very well cited paper in the future as it brings together a fundamental idea in plant evolutionary ecology (Baker's Rule) with invasion biology and agricultural pollination biology. The authors are to be congratulated.

Author's Response to Decision Letter for (RSPB-2019-2096.R0)

See Appendix B.

Decision letter (RSPB-2019-2096.R1)

23-Oct-2019

Dear Dr Brown

I am pleased to inform you that your manuscript entitled "Global scale drivers of crop visitor diversity and the historical development of agriculture" has been accepted for publication in Proceedings B.

Open Access

Paper charges

Sincerely,

Appendix A

Editor

Editor comment: The second reviewer felt that the data may be inadequate and that the methods and data handling were not sufficiently transparent. I agree with this reviewer about the backwards presentation of the accumulation curves. These curves should be used to assess whether the reported bee diversity in different regions is based on sufficient sampling. Cursory sampling of a crop outside its region of origin would lead to incorrect conclusions if contrasted with thorough sampling in the centre of origin. To be fair, you did include number of sampling localities in the models, but this is not really the same as using accumulation curves to assess whether sampling was sufficient or not.

Our response: We agree that cursory sampling of a crop outside its region of origin would lead to incorrect conclusions if contrasted with thorough sampling in the centre of origin. We have now assessed whether crops were more thoroughly sampled in the centre of crop or family origin and placed this in the supplementary document. Since we treated each crop in each study as an observation in regression modelling, the thoroughness of sampling for each crop is the number of sites and hours over which it was surveyed across all studies in that region. We have now assessed sampling thoroughness using comparisons of the mean and total number of survey locations and hours across all crops (Table 1 below) or for each crop (Figure 1 below). Across all crops, average and total survey sites and hours were greater when crops were grown outside their regions of origin, providing no evidence of sampling bias leading to inflated crop-visiting bee diversity in the centre of crop origin (Table 1). Average number of sites surveyed was 1.3 times greater in the region of family origin, but total number of sites surveyed was 2 times greater when crops were grown outside the region of their family's origin (Table 1). Similarly, Figure 1 indicates that only eight of 27 crops (30%) were surveyed at more locations in their realm of origin compared to outside their realm of origin, and these were equally distributed between crops of New and Old World origin (i.e. no evidence of geographic bias in survey effort). Figure 2 indicates that 12 of 27 crops (44%) were surveyed at more sites in their realm of family origin, which also provides no evidence of geographic bias. (We did not produce a similar figure for number of survey hours because only 22% of studies reported survey hours). In conclusion, this assessment indicates that there is no sampling bias of the kind that would provide an alternative explanation for crop and family origin effects that we document.

Unfortunately, the way most studies are reported does not allow the construction of accumulation curves of the kind suggested. This is because identity of bees visiting the crop in each study was almost always aggregated across sites and time periods within the study (i.e. the data reported in each study were the identities of bees visiting crops across all sites and time periods, not the identities of bees visiting crops at each site and time period). However, we believe the analysis presented in the previous paragraph shows there was no sampling bias favouring crops grown in their regions of origin, because the methodological differences that could give rise to such bias were not present.

Table 1: showing average and total number of survey sites and survey hours across all crops grown outside their region of origin (away) and in their region of origin (home) and family origin. The total number of studies in these categories is also shown (note that total number of studies does not add to 317 as some studies included multiple crops).

		Crop origin		Family origin	
		Away	Home	Away	Home
No. survey sites	Average	4.36	3.87	3.88	4.86
	Total	963	553	1006	510
No. survey hours	Average	50.84	47.53	48.22	53.62
	Total	2593	1426	2893	1126
No. studies	Total	221	143	259	105

Figure 1: Average number of locations surveyed in region(s) of crop origin (Home) compared to region(s) of introduction (Away). Green panel indicates New World crop (Nearctic and/or Neotropic), yellow panel indicates Old World Crop (Palearctic, Indomalaya, and/or Afrotropic). Blue regression line indicates great number of locations surveyed in region of origin, red line indicates greater number of locations surveyed outside region of origin, black line indicates no difference (note that these differences are not necessarily statistically significant).

Figure 2: Average number of locations surveyed in region(s) of family origin (Home) compared to region(s) outside of family origin (Away). Green panel indicates New World crop (Nearctic and/or Neotropic), yellow panel indicates Old World Crop (Palearctic, Indomalaya, and/or Afrotropic). Blue regression line indicates great number of locations surveyed in region of origin, red line indicates greater number of locations surveyed outside region of origin, black line indicates no difference (note that these differences are not necessarily statistically significant).

Editor comment: I also agree that Fig 3 is very poorly designed. I suggest putting Original realm and Introduced realm on the x axis and showing the actual means and errors connected by lines.

Our response: we have produced an improved Figure 3 (see below) that we believe also addresses Reviewer 2's concerns about the lack of presentation of regression coefficients. We have used R package jtools to produce a Forest plot (which is now commonly used outside of the meta-analysis context) showing the regression coefficients and 95% confidence intervals for all predictors in the model used for inference (the previous Fig 3 was produced manually in Excel and we can now see that there was a problem with the y-axis). We believe this new approach is appropriate as it presents effect sizes with a measure of precision (95% confidence intervals) that can also be used to assess statistical significance (1).

Figure 3: Increase in number of bee genera visiting crops when grown in the realm of crop origin and realm of crop family origin compared to all other realms, as well as Afrotropic, Palearctic, Nearctic, and Neotropic realms compared to the Indomalayan realm (Indomalay was used as reference as it exhibited the lowest average number of crop-visiting bee genera). Parameter estimates are derived from the top-ranked regression model (Table S2), with 95% confidence intervals, back-transformed (exponentiated) for ease of interpretation. Note that effects of predictors on the response are multiplicative (not additive) as a log-link function is used in negative binomial models, such that estimates with confidence intervals overlapping 1 (dashed line) are considered non-significant (at $\alpha = 0.05$).

Editor comment: I felt that you need to place this study in the broader context of what happens to pollination systems when species are introduced to new regions. There is a whole body of theory going back to Herbert Baker and others that is completely missed here. From line 62 you relate your question to the enemy release hypothesis, but you miss out on the rich literature about pollination of plants introduced outside their native ranges, especially Baker’s argument that this problem can be so severe that plants can only colonize new regions if they are capable of uniparental reproduction. To be sure Baker was mainly concerned with mate limitation, but he and others (notably Grant) also raised issues about limited pollinator availability outside the native range. The artificial selection that favours generalized pollination systems in crops is probably similar to the natural selection that favours generalized pollination systems in colonizing species. I would suggest consulting the literature on pollination of invasive plants, such as the reviews by David Richardson, Mark van Kleunen, John Pannell and others.

Our response: thank you for the suggestion, we have now re-written the introduction to incorporate this literature.

Editor comment: The Introduction would be more effective if it identified some specific predictions at the end after giving some prior rationale. In any case your two headings for rationale seem rather synonymous- “A home ground advantage” seems rather too similar to “The benefit of staying close to

the family home". Perhaps you should make it clearer that you mean family in the taxonomic sense. Change to something like "the benefit of being surrounded by relatives"?

Our response: thanks, we have now presented specific predictions after giving prior rationale, and have amended the second heading as suggested.

Editor comment: Line 269 Please formally report the % of crops that show this effect (a decrease in bee diversity outside their native range) and break this down by realm. It is conceivable for example that if most crops originated in the Americas and that this region has a high bee diversity and that many are grown outside this region in areas of low bee diversity that the whole study is simply an artifact of high bee diversity in the place where most crops originate and not a general phenomenon. Can you show that effects are reciprocal, ie that the same patterns apply for introductions from New to Old World and Old to New world? By giving the % of crops that show the effect, the reader could judge whether or not the overall effect is due to a few crops that show pronounced differences.

Our response: The figure below has been added to the supplementary material, and a paragraph now refers to it in the results section of the main manuscript. It shows the difference in mean number of bee genera observed visiting each crop when grown in its region of origin and outside this region. 20 of the 27 crops (74%) show some indication of greater crop visitor diversity when grown in the region of origin, with similar numbers of these crops originating in the New World (n = 9) and Old World (n = 11), showing that the effects are indeed reciprocal. This figure was produced using a regression model identical to the one presented in the paper (i.e. negative binomial model with all the same predictor variables), but with crop species treated as a fixed rather than random effect (i.e. using a GLM rather than GLMM), and an interaction between crop and crop origin (note that this interaction was not significant).

We would prefer to use the GLMM as we believe crop should be treated as a random effect because we have not included every possible crop in this study (data were not available for every crop). To show that the effects are reciprocal using the GLMM we have also tested for an interaction between the effects of crop origin and biogeographic realm and presented this in the supplementary material. A significant interaction would indicate that effects of crop origin depend on the biogeographic realm where observations were made, which is equivalent to saying that only crops originating in certain realms are visited by fewer bee genera outside their realm of origin (because crop origin for a given crop was 1 when the crop was surveyed in its realm of origin and 0 when it was surveyed in any other realm). The interaction was not significant, indicating the crop origin effect is general.

Average number of bee genera visiting each crop in region(s) of origin (Home) compared to region(s) of introduction (Away). Green panel indicates New World crop (Nearctic and/or Neotropic), yellow panel indicates Old World Crop (Palearctic, Indomalaya, and/or Afrotropic). Blue regression line indicates great visitation in region of origin, red line indicates greater visitation outside region of origin, black line indicates no difference (note that these differences are not necessarily statistically significant).

Other comments

Lines 132 -189 This section reads like a list and should be placed in a table, either in the main section or supplementary

Our response: most of this section has now been removed from the main text and placed in a table (Table S4).

Line 195 You chose to analyse bee genera, but there seems no reason not to also analyse bee species richness or other higher level groupings.

Our response: we chose not to analyse species richness because we believed the risk of species misidentification across such a broad range of studies was too high, and because not all studies identified bees to species level such that focusing on genus allowed us to include more studies and increase sample size. We chose not to analyse higher-level groupings because as one moves to higher-level groupings the likelihood of any pattern being present diminishes. We make this argument in the third paragraph under the subheading “design” in the materials and methods.

Line 205 How did you choose this particular mean-variance relationship. I assume you mean to control for overdispersion in a Poisson model or do you mean that the quadratic term best accounted for overdispersion in the default negative binomial model?.

Our response: We apologise for failing to make this clear in the text. Yes, negative binomial models were chosen as Poisson models were overdispersed, and the negative binomial model with a quadratic mean-variance relationship was a better fit than a negative binomial model with a linear mean-variance relationship. The text has been amended to read “Poisson models were over-dispersed, so negative binomial models were used, with a quadratic rather than linear mean-variance relationship specified as it provided a better fit to the data.”

Referencing format needs to be updated to the Proc B style. This also applies to the boxes and some other elements as well.

Our response: this has now been updated.

Line 47 Make it clear that you are referring to crops and not plants in general

Our response: amended.

Lines 50-53 This sentence is incomprehensible and needs to be re-written. What does this value of 12-13% refer to? In a place such as Europe is there a non-agricultural landscape? Support in what sense – forage or nesting sites or both?

Our response: this sentence and the paragraph containing it have been re-written.

Line 274 Sentence incomplete – more bee genera when.....family originated than when grown outside this realm

Our response: this sentence has been amended.

Line 275 reword the number of bee genera visiting crops within studies declined. Delete “within studies” throughout the sentence

Our response: this sentence has been amended.

Fig 4 should be modified so that it also makes sense in greyscale.

Our response: this figure has been amended.

Reviewer(s)' Comments to Author:

Referee: 1

Comments to the Author(s)

I see no mention of buzz pollination by specialist bees (bumble bees etc.) in angiosperm crops having poricidal anthers. Tomato is an example. Honey bees are incapable of sonicating tomatoes and other similar crops (blueberry, cranberry, chile peppers, eggplant, kiwi fruit). This has been reviewed by S. Buchmann and others.

Our response: the following sentence has been added to the introduction “Some crops require specialized forms of pollination, such as tomatoes and blueberries that are pollinated by the subset of bee taxa capable of vibrating anthers to release pollen (2, 3).”

One issue I have is that the 317 papers only contained data on 27 crops plants around the world. That seems a bit odd. I would like to see the authors mention that there are often specialist bees on certain crop plants. They barely mention this fact. For example, in the New World, the genus *Cucurbita* is pollinated by a guild of specialist bees in the genera *Xenoglossa* and *Peponapis*. These associations originated in Mexico and the bees followed these domesticated crops as they moved north out of Mexico. Similarly, the bee genus *Diadasia* specializes on mallows, cacti and sunflowers. The sunflower bee *Diadasia enavata* in California is an example of such specialization.

Our response: we acknowledge that there is a larger literature describing the bee taxa visiting a greater range of crops. We designed this study to test a specific set of hypotheses and therefore required a particular design that precluded the use of observations (and therefore crops) that did not meet the design criteria. Specifically, we tested whether crops were visited by fewer bee taxa when grown outside their region of origin or family's region of origin. Therefore, for a crop to be included in our study required observations of flower-visiting bees in its realm of origin and at least one other biogeographic realm. Further, since plant families can vary in their degree of specialisation (e.g. the typically more specialised floral forms of Solanaceae compared to the typically more generalised floral forms of Asteraceae), we attempted to avoid confounding plant family with region of origin by selected only plant families that include more than one crop species, with at least one crop species originating in the New World and one in the Old World. These design elements are described in the methods section.

Regarding the fact that there are often specialist bees on certain crop plants, the following lines appear in the introduction of our manuscript “...host switching has been observed in highly specialized bees, such as New World *Peponapis* (Cucurbitaceae specialists) and *Diadasia* (Asteraceae or Malvaceae specialists) that readily visit introduced Old World crops as long as they come from their preferred host families.”

You seemed to have missed citing the ground-breaking two volume book edited by David Roubik (STRI) and published by the FAO. It was published recently. In an earlier version, Roubik edited an FAO volume that had a table of crops grown around the world and listed their pollinators. It listed 1400 crop plant species. I am not sure why the articles you found only represent 27 crops. That seems very low. This might require a few sentences of explanation in the text. Please, please cite this ground-breaking work.

I have attached a pdf file of the first volume for your consideration.

Our response: Thank you, we have now cited this in the revised introduction

Regarding the 27 crops used in the present manuscript, this again is a consequence of the design requirements for hypothesis testing (see our response to the concern above).

Congratulations on a fine paper. I especially like Figure 2.

Our response: Thank you.

Referee: 2

Comments to the Author(s)

This manuscript explores how the diversity of pollinator services (bee genera) to crops change when crops are grown outside of the range they (or their progenitors or family members) are native to. While I found this a provocative and interesting question, I found the paper fell short of meeting its goals. One of the main concerns is that the sampling may be inadequate to answer the question and the presentation of the data and analyses lacked the transparency and specificity that would allow the reader to assess it.

I detail my main concerns here:

1. Although the paper's focus is bee genera, the names of these are never presented. Because the differences are relatively small (1 vs 4 genera) I was left wondering if the 1 was *Apis* thus reflected only visitation by cultivated species in many nonnative settings. The breadth and depth of the bee genera needs to be much better explicated.

Our response: We agree that in the original draft our focus on the global patterns came at the cost of some of the detail, such as regarding the bee taxa involved. We are pleased now to have rectified this by including a table of all the genera detected on crops across more than one realm (table S5) and another table of all the genera detected on crops in only one realm (table S6). These two tables help the reader to see which genera are contributing to the overall effects indicated in figures 2 and 3.

Regarding *Apis* in non-native settings, we too were curious about this. *Apis* occurred in all realms (which you can now see clearly in table S5) and visited all crops, and was present in most studies, therefore it did not contribute to patterns of difference. We performed the analysis without *Apis* and found that the effects of crop and family origin were slightly stronger when *Apis* was removed.

2. The assessment of power (accumulation curves) came second, after the main regression analyses. This seems backwards to me as I would imagine readers would appreciate understanding the rigor of the data first. I also thought that while the authors attempted to determine whether the number of crops sampled was enough, it was also important to know whether the sampling at a given crop was sufficient to capture all the diversity of visiting bees at that crop. The effort or intensity of pollinator sampling at this level was not addressed. It is possible that many bee taxa were missed but it is not possible to know with the data presented.

Our response: Some of these issues are already addressed in our response to the editor's comments above. We have now presented the accumulation curves before the regression analysis in keeping with this reviewer's advice

We recognise that it is possible that bee taxa were missed in some studies, which would be a problem for interpreting our analysis if there were biases in sampling intensity such that crops tended to be sampled more intensely in their realms of origin and family origin. However, we have now assessed sampling intensity and found that no such bias is present (see response to Editor's comments above).

3. The presentation of the (regression coefficients, etc) statistical analyses was lacking from the main text and the details were not made clear in the supplement. I found this a serious concern for a reader attempting to assess the importance of any claim made in the paper.

Our response: this has now been corrected by revising Figure 3 to include parameter estimates (with 95% confidence intervals) for all regression coefficients present in the best model (i.e. the model used for inference) in a Forest plot format.

4. I wondered about aspects of cropping systems that could affect pollinator diversity that differ in different parts of the world—large monoculture ag versus smaller or inter cropping systems. This should be addressed.

Our response: If some biogeographic realms are dominated by monoculture while others are dominated by small/inter-cropping systems, the biogeographic realm factor we included in modelling would account for this. At smaller spatial scales, this could be an important component of the unexplained variation between studies, which unfortunately was impossible to include in modelling as information about the form of agriculture was absent from most studies. This is now addressed in the second last paragraph of the discussion.

5. The assemblage effects section (starting line 290) was not mentioned earlier and it was not clear what 'networks' were being compared.

Our response: we have amended the text in the following ways. The introduction now includes a third sub-section providing a rationale for the prediction that suites of native crops will support more taxa than suites of exotic crops within each region. The 'methods' section has also been changed accordingly, with the first sentence of the paragraph describing genus accumulation curves now reading "Our third prediction – that suites of crops native to a region collectively support more

bee taxa compared to exotic crops – was evaluated by comparing bee genus accumulation curves for native crops and exotic crops within each biogeographic realm.”

6. Figure 3 was not very helpful. Figures that described the bee taxa and their distributions would be much more helpful.

Our response: Figure 3 has been revised as described (in response to the third concern above), and the new tables S5 and S6 detail the bee taxa and which realms they were detected in.

References

1. Nakagawa S, Cuthill IC. Effect size, confidence interval and statistical significance: a practical guide for biologists. *Biological reviews*. 2007;82(4):591-605.
2. Buchmann SL. Buzz pollination in angiosperms. *Buzz pollination in angiosperms*. 1983:73-113.
3. Roubik D. The pollination of cultivated plants. *A compendium for practitioners Vols*. 2018;1.

Appendix B

Response to referee

Referee: 3

Comments to the Author(s).

I think that the paper addresses a really interesting question in a thorough and systematic way. The previous round of reviews was very thorough and pointed out some significant limitations to the presentation of the work and how it was analysed. The manuscript as it stands looks fine to me and I can see that the authors have responded to the comments in detail. I therefore have no further requirements for changes, except that I spotted two typos:

Line 107 – “possess”

Line 251 – “actual”

Our response: we have now corrected these typos (see ‘Tracked Changes’ document).